# RelRoll: A Relative Elicitation Mechanism for Scoring Annotation with A Case Study on Speech Emotion

Yijun Zhou*
The University of Tokyo

Jinhong Lu†
University of Edinburgh

Xiang 'Anthony' Chen‡
University of California, Los Angeles

Chia-Ming Chang§
The University of Tokyo

Takeo Igarashi¶
The University of Tokyo

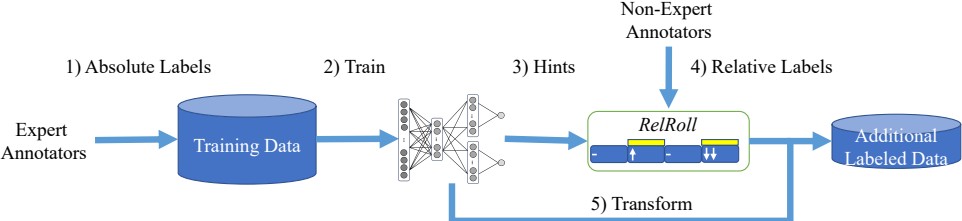

Figure 1: Concept of *RelRoll*.
1) Expert annotators first give reliable absolute labels to *training data*. 2) The system trains a network using the dataset. 3) The network provides hints (highlights) for non-expert annotators. 4) Non-expert annotators give relative labels to new data. 5) Finally, the network transforms relative labels to absolute labels and additional labeled data is acquired.

## ABSTRACT

It is challenging for non-experts to give reliable labels in scoring annotation tasks because of the lack of domain knowledge and the high variability. We present *RelRoll*, a mechanism allowing non-experts to give reliable ratings, with a case study on speech emotion. It includes two main features: a relative labeling interface highlighting emotion-changing sentences and an approach to estimating absolute labels from relative labels. Highlighting can help non-experts focus on sentences needing their actions. Given new sentences, we utilize a network to predict emotion-changing sentences and highlight them on the interface, which is trained on *training set* labeled by experts' absolute annotations. Non-experts will give relative labels to new sentences through our interface. We estimate the absolute labels of new sentences by combining the network output and relative labels. We ran a user study to compare the proposed interface to the commonly used absolute interface. Results showed that *RelRoll* could improve the annotation agreement and the absolute label annotation accuracy. We discussed the user experience enhancement on helpfulness, intuitiveness, and easiness.

**Index Terms:** Human-centered computing—Interaction design—Interaction design process and methods—User interface design; Information systems—Information retrieval—Retrieval tasks and goals—Sentiment analysis

## 1 INTRODUCTION

Scoring annotation is ubiquitous (image, audio, motion, etc.) but challenging, such as image quality assessment, speech emotion anno-

*e-mail: yijun-z@g.ecc.u-tokyo.ac.jp

†e-mail: jinhong.l@ed.ac.uk

‡e-mail: xac@ucla.edu

§e-mail: info@chiamingchang.com

¶e-mail: takeo@acm.org

tation, rating physical mobility score of clinical videos, etc. Ideally, these tasks would be conducted in a laboratory and trained experts with specific domain knowledge are recruited to finish the task under supervision. However, it is always time-consuming, costly, and limited in the number of annotators. To enable more efficient and more cost-effective annotation collection, it is a desirable option to include non-expert annotators (e.g., crowd workers) in the loop. Here we define a non-expert annotator as other works, a person (1) has no prior annotation experience in a specific domain and (2) has no sufficient domain knowledge [9, 21]. For example, it is not feasible to have a crowd worker who has never learned emotions and has no annotation experience to understand the full range of emotion scales and give reliable annotations [1].

We focus on the speech emotion annotation task in this study, which is important to Speech Emotion Recognition (SER) as an example of difficult scoring annotation tasks. However, we expect our techniques to be extensible to other tasks of scoring sequences of data. Most prior research uses a dimensional description model [40] to label emotions with numbers, which describes emotions along several orthogonal descriptive axes, such as pleasure (from unhappiness to happiness), arousal (from sleepiness to excitement), etc. The advantage of using the dimensional description model is that its dynamic characteristics correspond to the intrinsic property of emotions, continuously changing [12], which allows the smooth transition between emotional states. Due to a lack of expert annotators, while non-expert annotators are possible in other domains (e.g., labeling common objects in images), they would struggle to label emotions in speech because (1) it is infeasible for non-experts to learn the full range of relationships between absolute values and emotions and (2) the emotion annotation task is highly variable, which may introduce variations among non-experts.

In this study, we propose *RelRoll*, a mechanism allowing non-expert annotators to give reliable scoring labels through relative labels. Here we define absolute labels as continuous numerical labels of even intervals (e.g., "1", "2", "3", etc.), and discrete relative labels as level-changing labels of uneven intervals (e.g., "higher"or "lower"). Instead of learning the full range relationships between numerical values and emotions, it is easier for non-experts to tell

how much the emotion changes roughly. For example, it is easier to tell the change in emotion of a sentence relative to the previous one (e.g., higher arousal) than to give its absolute arousal value (e.g., 4). The concept of *RelRoll* is shown in Fig. 1.

First, we train a neural network on *training set* annotated by experts through absolute labels, which predicts the absolute emotion (pleasure and arousal) labels. We utilize the network to predict differences between contiguous sentences in a dialogue from *test set* and set a threshold to determine which sentences will be highlighted on the interface. Highlighted sentences are supposed to have emotional changes, which we assume annotators shall focus on during the labeling process. Non-expert annotators are asked to give relative labels to sentences compared to their previous sentences while listening to the dialogue. We compute the latent emotion values of those sentences from relative labels by *Bradley-Terry Model* [4]. We then estimate the absolute labels of those sentences by the affine transformation of the latent emotion values to the distribution predicted by the network. We ran a user study to compare the proposed relative labeling interface to the traditional absolute (non-relative) labeling interface in a speech emotion annotation task. Results demonstrated that the annotation agreement of the proposed interface with highlights was higher than both the relative interface without highlights and the absolute interface. In addition, it showed that our interface could improve the annotation accuracy. Furthermore, participants responded with their preference for our proposed interface over the traditional interface, on *easiness*, *intuitiveness*, and *helpfulness* in questionnaires. Their responses proved that *RelRoll* could lower the learning curve of annotation.

Our contribution is mainly three-fold:

- *RelRoll*, an approach to estimating absolute labels from relative labels given by non-expert annotators, which can lower the cost of collecting labels for large datasets.

- a relative labeling interface design for non-expert annotators in scoring annotation tasks, which can improve annotation agreement, labeling accuracy, and user experience.

- takeaways from a user study eliciting absolute labels from relative labels in a speech emotion annotation task, validating the usability of *RelRoll*.

## 2 RELATED WORK

In this section, we explain the background and describe related work. We start with labeling tasks (scoring annotation and speech emotion) and then introduce labeling mechanisms (absolute and relative labeling).

### 2.1 Scoring Annotation and Speech Emotion

Many scoring annotation tasks are of high variability, most of which are done in laboratory and controlled conditions. Speech emotion annotation usually involves multiple annotators to account for variations in emotion perception across individuals [7]. Image quality assessment is designed for evaluating the human perceptual quality of images, which needs subjective annotations [20]. In addition, the rating is also commonly used in clinical diagnosis of physical mobility [34], where experts rate the severity of patients' mobility dysfunctions and recognize their need for supervision or assistance.

In speech emotion annotation, there are two principal models to describe emotions, the categorical description model and the dimensional description model. The categorical description model [16] is more common, in which emotions are described around some basic emotion categories, such as happy, angry, sad, etc. The other commonly used dimensional model is developed by Mehrabian and Russel [35], using pleasure, arousal, and dominance to represent all emotions. Pleasure and arousal are termed as "core affect", while

dominance is related to the relationship between people. We only focus on pleasure and arousal in this work.

Semi-natural or natural datasets are the most suitable for dimensional emotion analysis [19], such as IEMOCAP [7], MSP-PODCAST [33]. In these datasets, emotional speech is collected from talk shows, interviews, panel discussions, dialogues, and plays within a context. Sentences are collected from a continuous speech, which inspired our design of a continuum labeling interface. In this work, we use the improvised sessions (80 dialogues) from IEMOCAP following the suggestions from [47], as they are more similar to natural speech and more emotion-balanced [8].

### 2.2 Absolute and Relative Labeling

Labeling tasks are commonly done through absolute labeling, e.g., giving a numerical or semantical label to an instance. It has wide applications in video, image, audio, text, 3D model, and other data formats [9, 18, 22, 29, 38, 45]. However, it is not always the most reliable method. When there is a lack of commonly accepted scale, absolute labeling will introduce disagreement and inconsistency to the result [6]. Especially when non-experts are involved, the disagreement and inconsistency will invalidate the labels. As for classification annotation, many techniques including structured labeling [10, 27], explanatory debugging [28], etc., tried to improve the consistency and agreement among crowdsourcing workers.

An alternative method to absolute labeling is relative labeling. Relative labeling means that annotators will compare one instance to another, through a preference or ranking expression, such as "better than", "worse than", "equal", etc. This method focuses on the local relationship instead of the global scale, which makes the task more intuitive and easier. Relative labeling is reliable and accurate in many tasks when variability is involved in the task, for example, emotion labeling, age labeling, quality labeling, and so on [15, 24, 31, 32, 37]. However, collecting relative labels is costly as a pairwise comparison will square the number of labeling times (i.e. $N$ pairwise comparisons per instance, $N^2$ labels needed in total). Many researchers try to improve this challenge through interaction support. SorTable [41] allows users to make sets of relative labels and uses the Bayesian method to estimate the complete ranking. Goldilocks [11] displays previously annotated items as anchors and allows users to give the upper and the lower bounds of a new instance by referring to already-labeled instances.

Annotation process can be integrated with techniques to realize more efficient labeling. OLALA [42] only selects the ambiguous predicted portions for humans to annotate through active learning. Felix *et al.* [18] proposed an unsupervised machine learning method to allow users to discover and create label categories in an evolutionary fashion. Ranjay *et al.* [26] proposed a technique for producing rapid absolute labels for a running stream of images. SCRIBE [30] introduced an approach to collecting complete transcripts of videos by merging partial inputs from groups of non-expert captionists in real time. SCRIBE can be effective for obtaining relatively accurate answers from a group of non-experts, but it is infeasible for only one non-expert in most cases.

In the emotion labeling task, Self-Assessment Manikin (SAM) [3] is the most commonly used interface, which can be categorized as an absolute labeling interface. Two-dimension slider with color representation [13], facial expression suggestion [5], and mobile integration [46] have also been introduced to assist users with labeling emotions. However, none of the research tried to include non-expert annotators because uncertainty may be introduced. To allow non-expert annotators to give reliable labels, we select relative labeling as our approach. To balance the uncertainty and cost, we integrate the *Bradley-Terry Model* [4] with a continuum labeling interface, which allows us to obtain $N^2$ pairwise difference labels in $N$ time.

## 3 RELROLL INTERFACE

In speech emotion annotation tasks, non-experts may be faced with a steep learning curve to learn the relationship between speech emotions and numerical values through tutorials and practice. We need to reduce the difficulty of labeling tasks to include non-experts in the loop. Another challenge is the low agreement among annotators due to the high variations in emotion understanding, which also exists among expert annotators. Thus it is also important to improve the agreement among the annotators to ensure the labeling quality. In this section, we explain the details of designing the interface to elicit relative labels from non-experts, with two key design goals, (1) to lower the learning curve and (2) to reduce the variability. We compared the proposed interface to the baseline interface with design details.

### 3.1 Baseline Interface

The most commonly used labeling interface in speech emotion annotation tasks is the absolute labeling interface. Fig. 2 shows an example of an absolute labeling interface for emotional speech. We created this interface by modifying the prior work, SAM [3]. It resembles the commonly used speech emotion labeling interface, which also serves as the baseline interface in our user study. It has three main components, the brief description of the dialogue content, the labeling component, and the transcript. Users have to listen to a sentence (one blue block in Fig. 2) and give their label to the sentence by a slider. As in this example, the numerical scale is 1, 2, 3, 4, and 5, which is used in the data collection process of IEMOCAP [7]. This type of interface does not support lowering the learning curve and reducing variability. For example, it is difficult to assign an absolute numeric value to pleasure for a non-expert, as they do not understand the relationship between those numbers and pleasure. We add the following two features (not present in the original SAM) to the baseline to facilitate fast labeling.

*Non-stop playing with sentence-level splits.* The labeling component also serves as an audio streaming player of the dialogue. In each task, users are required to focus on one emotional dimension (pleasure or arousal) of one speaker in the dialogue. During labeling, the playing head indicates which sentence is playing. We choose the non-stop playing for fast labeling, which requires users to give labels before one sentence ends. Users can modify their labels by rewinding to previous sentences if needed.

*Keyboard interaction.* Users operate labeling and audio playing through keyboard interactions, up arrow and down arrow to move the thumb on the slider, left arrow and right arrow for the rewinding and fast-forwarding of the audio. We choose the keyboard interaction instead of the mouse interaction, as we believe it is easier to operate.

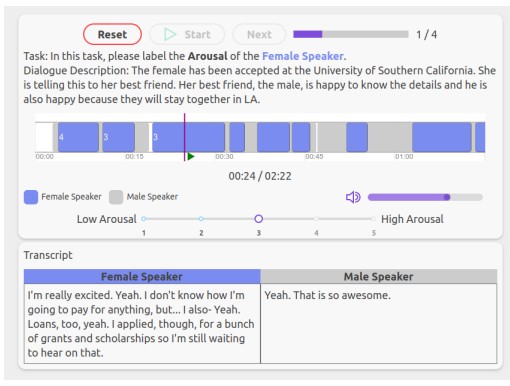

Figure 2: Baseline interface.
In this example, users are listening to the 3rd sentence and giving a label, 3.

### 3.2 Proposed Interface

Our proposed interface of *RelRoll* is shown in Fig. 3. It has the same components as in the baseline interface. It also shares *non-stop playing with sentence-level splits* and *keyboard interaction* as the baseline interface. Three main different features are described below.

*Relative labeling.* We use a five-level scale of relative labels, "much higher": ↑↑, "higher": ↑, "no difference": —, "lower": ↓, and " much lower": ↓↓. They allow users to respond how much they perceive a sentence's pleasure or arousal is different from its previous one's. We choose five as the number of levels because it is readily comprehensible and enables people to express their perception of emotion changes. Users give relative labels by pressing up arrow and down arrow on the keyboard. If no action is taken when a sentence playing ends, there will be a dash — for "no difference" shown.

*Highlight suggestions.* To further assist non-expert annotators, we highlight sentences that may need them to take action. We train a neural network to predict which sentence has a different emotion as compared to its previous one. If the network predicts a difference, there will be a light yellow highlight above the blue sentence block. We believe highlights can make labeling easier for non-experts, as highlights will reduce their mental effort.

*Transcript reference.* Another feature implementing the idea of relativity is the transcript component. Transcripts of all speakers are shown below the labeling component, where the last sentences are also shown to users for reference. We believe that transcripts of last sentences can remind users how last sentences sound. It may help them make decisions on labeling emotion changes between current sentences and previous sentences.

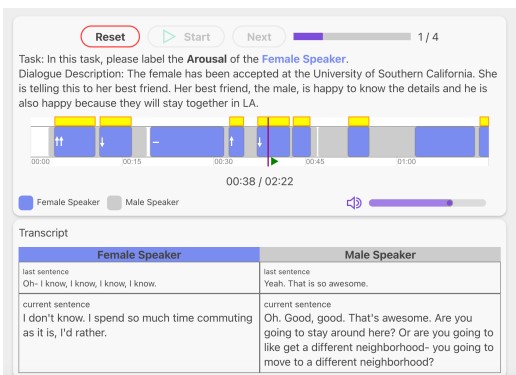

Figure 3: Interface of *RelRoll*.
In this example, users are listening to the 5th sentence and giving a relative label, ↓.

## 4 IMPLEMENTATION

An overview of the mechanism is shown in Fig. 1. We use an existing dataset (IEMOCAP) with expert labels as *training set*, and then train the speech emotion prediction network (Sect. 4.1). The network is used for highlighting sentences for non-expert annotation and calculating the distribution of the sentences' emotion. The relative labels from non-experts are quantified to absolute labels using a Bradley-Terry Model (Sect. 4.2) and affine transformation of the distribution (Sect. 4.3).

### 4.1 Network

The SER neural network in the mechanism is designed for two purposes: (1) to predict emotion-changing sentences (highlighted sentences in Fig. 3) on the interface and (2) to predict the emotion value distribution of new sentences of a speaker from a conversation.

In this section, we describe features, architecture, and how we predict highlighting.

*Features.* We extract frame-by-frame low-level descriptors (LLDs) such as fundamental frequency and Mel-frequency cepstral coefficients (MFCCs) by a commonly used audio processing tool, Opensmile [17]. For each sentence, we extract global statistics of the LLDs, such as arithmetic mean (e.g., mean of the fundamental frequency). These global statistics are referred to as high-level functionals (HLF). This method generates 6,373 features for each sentence.

*Network architecture.* To predict emotion difference between a pair of sentences, we build a simple SER neural network similar to the work by Parthasarathy*et al.* [36]. The SER neural network takes the input into the shared hidden layers and then separately connects them for pleasure difference and arousal prediction. The separated layer can learn dimension-dependent information and can optimize for its own goal. The architecture of the SER neural network is shown in Fig. 4. The depth of the shared hidden layer in the model is 4 and the dimension of each layer is [2048, 512, 64, 64]. There is a rectified linear unit (ReLu) activation function in between the shared hidden layers for introducing non-linearity into the output. Then the dimension of the separated hidden layer is 6. All of these hyper-parameters are decided by the preliminary experiments we conducted.

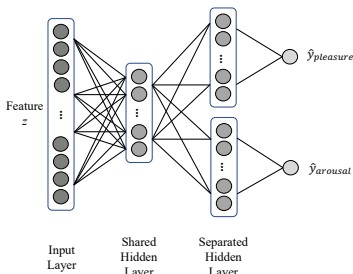

Figure 4: SER neural network.
The SER neural network takes the features ($z$) of one sentence as input, and predicts the emotion values ($\hat{y}_{pleasure}$ and $\hat{y}_{arousal}$).

The objective function of this task is to minimize the mean square error between the ground truth differences and the predicted differences. We denote the loss function of pleasure as following (arousal is same):

$$Loss_{pleasure} = \frac{1}{N}\sum(y_{pleasure} - \hat{y}_{pleasure})^2 \quad (1)$$

where $y_{pleasure}$ is the ground truth value of pleasure and $\hat{y}_{pleasure}$ is the value of pleasure predicted by the SER neural network.

We follow the optimization procedure in [36] and make it a balance of weighted pleasure difference and weighted arousal difference. The following formula describes the overall loss for the SER neural network:

$$Loss = \alpha \times Loss_{pleasure} + (1 - \alpha) \times Loss_{arousal} \quad (2)$$

where $\alpha$ adjusts weights of pleasure loss and arousal loss. We choose 0.55 as the value of $\alpha$ for the best performance after experimenting with different values. The training was conducted on a GPU machine with Pytorch version 1.5 by mini-batch training using Adam optimisation (learning rate 0.0002) [23], the batch size is 16, and the epoch is 30.

*Highlighting prediction.* As for predicting highlighted sentences, we set a threshold of predicted numerical difference as 0.5, above which the latter sentence of two consecutive sentences will be highlighted. Please note that we still choose to use mean square error

as the objective function, instead of directly using those loss functions designed for binary classification problems. This is because the second purpose of the network mentioned at the beginning of this subsection, which is we need the SER network to predict the distribution of new data.

### 4.2 Quantification of Relative Labels

In this section and Sect. 4.3, we take pleasure as the example, as arousal is processed in the same way.

Given $N$ sentences of one speaker in a conversation, we assume there is a single ranking $R$ of pleasure value by:

$$R = \{s_1 \succeq s_2 \succeq ... \succeq s_N\} \quad (3)$$

where sentence $s_1$ has a higher or equal pleasure value than sentence $s_2$, and so on. We use an extended Bradley-Terry Model [4] to describe the probability of this situation:

$$P(s_1 \succeq s_2 \succeq ... \succeq s_N | \pi) = \prod_{i=1}^{N} \frac{e^{\pi_i}}{\sum_{j=1}^{N} e^{\pi_j}} \quad (4)$$

where $\pi_i$ represents the latent pleasure value of $s_i$. It is trying to estimate $\pi$ by maximum likelihood. By using this method, we could get the latent pleasure values of those $N$ sentences from the relative labels of an annotator.

Due to the sequential comparisons acquired by a continuum labeling interface, we cannot acquire a full set of pairwise comparisons of the set. For example, if sentence $s_2$ is "lower" than sentence $s_1$ and sentence $s_3$ is "higher" than $s_2$, we cannot interpret the relationship between $s_1$ and $s_3$. Thus we add regularization to complement the full comparison set.

### 4.3 Estimation of Absolute Values

The final goal is to estimate the absolute values of pleasure $Y$ from the latent pleasure values $Y_L$ obtained in Sect. 4.2. Given sentences of one speaker in a conversation, we make an assumption that both the latent pleasure values and absolute pleasure values follow normal distributions.

We denote the latent pleasant values $Y_L$ of the sentences computed from Equation 4 as:

$$Y_L \sim \mathcal{N}(\mu_L, \sigma_L^2) \quad (5)$$

where $\mu_L$ is the mean of latent pleasure values, and $\sigma_L$ is the standard deviation.

We denote the distribution of absolute pleasure values $Y_P$ predicted by the SER neural network (the second purpose of the SER neural network described in the Sect. 4.1):

$$Y_P \sim \mathcal{N}(\mu_P, \sigma_P^2) \quad (6)$$

where $\mu_P$ is the mean of predicted absolute pleasure values by the SER neural network and $\sigma_P$ is the standard deviation.

Then we try to obtain the final estimation $Y$ of the absolute pleasure value by affine transforming $Y_L$ to make it follow the $Y_E$'s distribution. We denote the affine transformation as:

$$Y = AY_L + b \quad (7)$$

where $A$ can be calculated by $\sigma_P/\sigma_L$, and $b$ can be calculated by $\mu_P - A\mu_L$. After the affine transformation, we clip off the final absolute pleasure estimations $Y$ to the range of the scale used in IEMOCAP, i.e. from 1 to 5.

We evaluate the estimated absolute emotion values using the concordance correlation coefficient (CCC). CCC measures the agreement between the ground truth emotion values and the estimated emotion values. It is a commonly used reliable evaluation metric in

SER, as it is penalized in proportion to the deviation when predictions shift in value [39]. The range of CCC is from -1 to 1, with a perfect agreement at 1.

## 5   USER STUDY

We conducted a user study to examine the effectiveness of *RelRoll*. In particular, we compared three conditions (The main differences are shown in Fig. 5): (1) *RelRoll* with highlight suggestions (**Highlight**), the same as described in Sect. 3.2, (2) *RelRoll* without highlight suggestions (**Relative**), the interface described in Sect. 3.2 with highlight suggestions ablated, and (3) a traditional absolute labeling interface as a baseline (**Absolute**), same as described in Sect. 3.1. We aimed to answer these Research Questions (RQs) in this study:

**RQ1**: Would relative labels improve the annotation agreement (inter-annotator agreement), and would highlight suggestions further improve it?

**RQ2**: Would highlight suggestions and relative labeling improve the annotation accuracy? Would relative labels from non-expert annotators be valid and reliable?

**RQ3**: Compared with the traditional absolute labeling interface, would *RelRoll* be easier for non-expert annotators to use? Would it lower the learning curve of annotation?

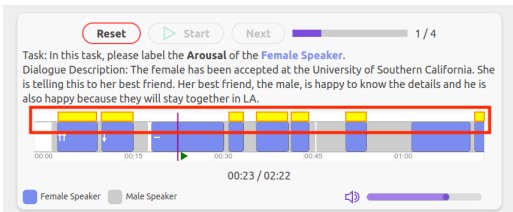

(a) **Highlight** condition.

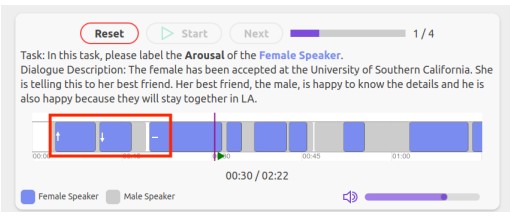

(b) **Relative** condition.

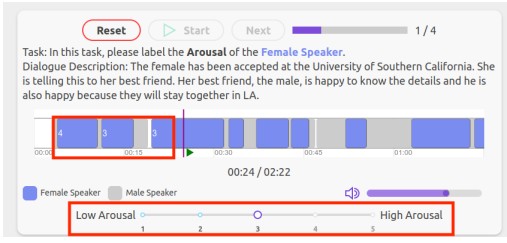

(c) **Absolute** condition.

Figure 5: Three conditions in the user study.
Red boxes reflect the differences among the three conditions. In (a) **Highlight** condition, the red box shows highlights suggesting emotion-changing sentences, the yellow bars above blue sentence blocks. In (b) **Relative** condition, the red box shows the relative labels users give. In (c) **Absolute** condition, the upper red box shows absolute labels users give; the lower red box shows the numerical slider operated by the keyboard.

Table 1: Demographics of user study participants.

Genders (self-identified) contains 'F' (female) and 'M' (male). Ethnicity contains "AA" (African-American), "AI" (Asian-Indian), "C" (Caucasian), "H" (Hispanic), and "NA" (Native-American).

| Condition | Gender F | M | Age (SD) | Ethnicity |
|---|---|---|---|---|
| Highlight | 6 | 6 | 34.17 (7.97) | AI(6), C(5), NA(1) |
| Relative | 5 | 7 | 41.17 (9.99) | AI(5), C(7) |
| Absolute | 5 | 7 | 39.08 (6.63) | AA(1), AI(3), C(7), H(1) |

### 5.1   Pilot Study

We ran a pilot study with three participants to check the interface design and the choice of dialogue. We tested keyboard operation and mouse operation, and received all preferences on keyboard operations. We tested four dialogues of different base emotions (anger, happiness, sadness, and neutral) with participants to check whether the base emotion can be a factor influencing the annotation results. While we found similar annotation accuracy results among different base emotions, participants reported fatigue when the dialogue was long in duration. Thus we considered the duration when we chose the dialogue for the user study, limiting it to 3 minutes or less. In addition, we found original sentence-level split meta information is not perfectly suitable for non-experts to annotate (e.g., there are short fillers such as *"Um. Okay."*). We filtered out sentences not suitable for the user study.

### 5.2   Participants

We recruited 36 participants through the Amazon Mechanical Turk Masters service [14] to realize a between-subject experiment design, where 12 participants were assigned to each condition. To guarantee all participants would be "non-experts", we asked them two questions in the pre-test questionnaire: (1) whether they had previous experience in labeling emotional data and (2) whether they had been trained on emotion comprehension to ensure that all of them were non-expert annotators. Only those who answered "no" to both questions could proceed with the experiment. We also required that they were native or fluent as native English speakers, to reduce the hinder on language. As for the operation on the keyboard, we required that they had access to a desktop or laptop with arrow keys on the keyboard to finish the experiment. We paid each participant 10 USD as compensation for their participation in the experiment. Participants included 16 females and 20 males from the USA and India, the mean age of whom was 38.14 years old (min=22, max=63, SD=8.81). Table 1 presents the detailed demographic information of each condition.

### 5.3   Procedure and Task

Participants completed the experiment through a web browser such as Chrome or Firefox. For 3 conditions, participants went through the same procedure, except for the interface differences. Each participant first watched a tutorial video of basic dimensional emotion concepts and an introduction to the interface (5 - 10 minutes). In addition, the video also showed examples of speech of different emotion values to help them better understand related concepts. Then, they were asked to practice labeling a short dialogue (2 - 5 minutes). Note that participants were allowed to watch the introduction videos and to practice multiple times. After practice, they proceeded to the labeling tasks (12 - 15 minutes). As we required users to focus on one speaker and one dimension in each task, there were four tasks, *Female-Pleasure*, *Female-Arousal*, *Male-Pleasure*, and *Male-Arousal*. To obtain unbiased feedback from non-expert annotators and to avoid order effects, we counter-balanced the order of four tasks. Finally, they answered a post-study questionnaire and

completed a background survey (15 minutes). The total time of the experiment is around 40 minutes.

We selected one dialogue used in the pilot study for the user study, which was 2 minutes 40 seconds containing fifteen sentences by Female Speaker and sixteen sentences by Male Speaker. The neural network predicted highlights above sentences in the **Highlight** condition. Here we assumed a mapping rule from absolute emotion value differences to relative labels as $\Delta emotion > 1.0 = \uparrow\uparrow$, $0.5 \leq \Delta emotion \leq 1.0 = \uparrow$, $-0.5 < \Delta emotion < 0.5 = -$, $-1.0 \leq \Delta emotion \leq 0.5 = \downarrow$, and $\Delta emotion < -1.0 = \downarrow\downarrow$. For example, if the latter sentence $s_1$'s ground truth pleasure value is 4.0 and the former sentence $s_0$ is 3.0, then it is mapped to a relative label, $\uparrow$ ($\Delta pleasure = 1.0$). F1 score was 0.914 for pleasure and 0.933 for arousal. Fig. 6 shows the confusion matrix of the highlight prediction results. *Predicted highlights* mean highlights acquired from network predictions, while *True highlights* mean highlights transformed from the ground truth. Both used the mapping rule stated above.

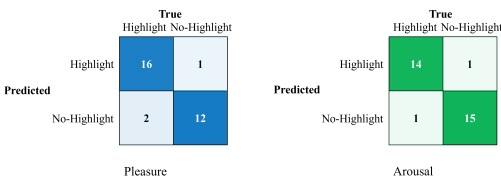

Figure 6: Confusion matrix of highlights prediction.

### 5.4 Measurement

#### 5.4.1 Annotation agreement

We evaluate the inter-annotator agreement by three metrics, percent agreement, Krippendorff's Alpha, and Fleiss' Kappa, following the previous work [2, 32]. Percent agreement is commonly used and intuitive, which is also suitable for conducting the *Two-Way ANOVA* and *Tukey HSD Test* at the sentence level to compare the agreement of different conditions. Both Krippendorff's alpha and Fleiss' kappa are reliability coefficients, taking into account the agreement expected by chance. They are obtained by taking all sentences and all annotations from each condition into the computation. Thus, there is no statistical significance test on either coefficient.

#### 5.4.2 Annotation accuracy

As for **Highlight** condition and **Relative** condition, we measures the relative annotation accuracy of each participant. Relative labels were treated as ordinal labels for computation, the accuracy of which was measured by Spearman's rank correlation coefficient ($\rho$) [44]. Spearman's rank correlation coefficient is suitable for measuring ordinal data that have uneven intervals, the range of which is from -1 to 1. CCC measures the absolute annotation accuracy in **Absolute** condition and estimated absolute values in **Highlight** condition and **Relative** condition. *Two-Way ANOVA* and *Tukey HSD Test* are conducted for both relative annotation accuracy and absolute annotation accuracy.

#### 5.4.3 Task completion time

We record the time usage of each task. The time usage refers to the time between participants click "Start" and "Next" button. We also count the times of participants press left arrow key for rewinding, which can also be an indicator of labeling speed and an inspector of labeling behaviors.

#### 5.4.4 Questionnaire on user experience

We use a five-scale Likert to ask participants to rate three factors of the interface: (1) Easiness, (2) Intuitiveness, and (3) Helpfulness. *Friedman's Test* and *Tukey HSD Test* are conducted on the rating

Table 2: Annotation reliability coefficients.

($\alpha$: Krippendorff's Alpha, $\kappa$: Fleiss' Kappa).

| | pleasure | | arousal | |
| Condition | $\alpha$ | $\kappa$ | $\alpha$ | $\kappa$ |
|---|---|---|---|---|
| Highlight | **0.574** | **0.229** | **0.708** | **0.467** |
| Relative | 0.487 | 0.203 | 0.706 | 0.446 |
| Absolute | 0.265 | 0.139 | 0.453 | 0.190 |

results. We also ask participants to give reasons of their ratings. In addition, we allow participants to give a free-form comments on the advantages and disadvantages of the interface, or to give any comments on their experience of any part of the experiment procedure.

### 5.5 Results

#### 5.5.1 Annotation agreement

Fig. 7 shows the results of inter-annotator agreement on percent agreement (P), and Table 2 shows the results of Krippendorff's Alpha ($\alpha$) and Fleiss' Kappa ($\kappa$). As for Krippendorff's Alpha, we use the 0.667 threshold as the standard of quality [25]. As for Fleiss' Kappa, we use the 0.400 threshold as the standard of quality [43].

*Percent Agreement.* **Highlight** shows an improvement as compared to **Relative** and **Absolute**. As for pleasure, there is a statistical significance in **Highlight** (P=68.5%) vs. **Relative** (P=57.1%) ($p < 0.05$), **Highlight** vs. **Absolute** (P=39.1%) ($p < 0.01$), and **Relative** vs. **Absolute** ($p < 0.01$). As for arousal, there is a statistical significance in **Highlight** (P=69.3%) vs. **Relative** (P=62.3%) ($p < 0.05$), **Highlight** vs. **Absolute** (P=41.0%) ($p < 0.01$), and **Relative** vs. **Absolute** ($p < 0.01$).

*Krippendorff's Alpha and Fleiss' Kappa.* As for pleasure, We do not find any condition exceeding the threshold on Krippendorff's Alpha or Fleiss' Kappa. However, **Highlight** shows an improvement of 17.8% of Alpha and 12.8% of Kappa as compared to **Relative**, and both **Highlight** and **Relative** show a great improvement than **Absolute**. As for arousal, we find both **Highlight** and **Relative** exceed the threshold on Krippendorff's Alpha and Fleiss' Kappa. In addition, **Highlight** shows an improvement of 0.8% in Alpha and 4.7% in Kappa as compared to **Relative**, and both **Highlight** and **Relative** show a great improvement over **Absolute**.

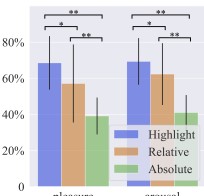

Figure 7: Annotation Percent Agreement.
(*: $p < 0.05$, **: $p < 0.01$).

#### 5.5.2 Annotation accuracy

Fig. 8a shows the relative annotation accuracy ($\rho$) results of **Highlight** condition and **Relative** condition. As for pleasure, **Highlight** ($\rho$=0.930, 95% CI=[0.899, 0.982]) is higher than **Relative** ($\rho$=0.764, 95% CI=[0.708, 0.820]). As for arousal, **Highlight** ($\rho$=0.962, 95% CI=[0.948, 0.977]) is higher than **Relative** ($\rho$=0.865, 95% CI=[0.831, 0.898]). Statistical significance has been found in both pleasure ($p < 0.01$) and arousal ($p < 0.01$).

Fig. 8b shows the results of the absolute annotation accuracy measured by CCC. As for pleasure, statistical significance is found

on **Highlight** (CCC=0.713, 95% CI=[0.694, 0.732]) vs. **Relative** (CCC=0.542, 95% CI=[0.491, 0.592]) ($p < 0.01$), **Highlight** vs. **Absolute** (CCC=0.231, 95% CI=[0.166, 0.297]) ($p < 0.01$), and **Relative** vs. **Absolute** ($p < 0.01$). As for arousal, statistical significance is found on **Highlight** (CCC=0.780, 95% CI=[0.760, 0.800]) vs. **Relative** (CCC=0.720, 95% CI=[0.695, 0.744]) ($p < 0.01$), **Highlight** vs. **Absolute** (CCC=0.539, 95% CI=[0.442, 0.636]) ($p < 0.01$), and **Relative** vs. **Absolute** ($p < 0.01$). Note that in **Absolute** condition, annotation accuracy is even lower than the network prediction performance. The accuracy of network prediction on both pleasure (CCC=0.522, the dotted line in Fig. 8b) and arousal (CCC=0.671, the dashed line in Fig. 8b).

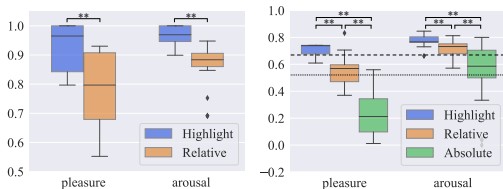

(a) Relative annotation accuracy (b) Absolute annotation accuracy
($\rho$). (CCC).

Figure 8: Annotation accuracy results.
In Fig. 8b, dotted line is the network prediction accuracy on pleasure, and dashed line is the network prediction accuracy on arousal (**: $p < 0.01$).

### 5.5.3 Task completion time

While participants labeled the same amounts of data across the three conditions, we find a learning effect in **Highlight** condition, as both time usage (average time (s) in Fig. 9a: 1st=178.3, 2nd=160.3, 3rd=156.1, 4th=152.0) and rewinding counts (average counts: 1st=3.33, 2nd=1.42, 3rd=0.92, 4th=0.58 in Fig. 9b) decreased as tasks went on. We haven't found the same behavior in **Relative** condition or **Absolute** condition. We give a discussion on it in Sect. 5.6.

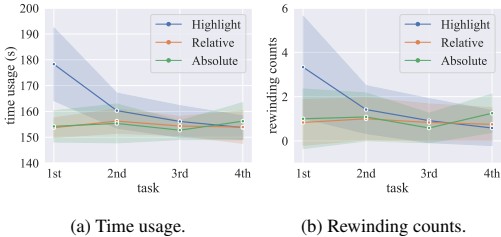

(a) Time usage.     (b) Rewinding counts.

Figure 9: Task completion results.

### 5.5.4 Questionnaire on user experience

Fig. 10 shows the results of participants' feedback on Easiness, Intuitiveness, and Helpfulness.

*Easiness.* There is a statistical significance in **Highlight** vs. **Absolute** ($p < 0.01$). **Participants mentioned the "reference" in interface design reducing the memory load in their reasons.** *P3 (Highlight): "It helped me because I could refer to the change in emotion from audio sequence to audio sequence just by looking at the arrow (up or down) on the previous audio clip that I played and labeled. Also, the transcript below reminds me how that sentence sounded." P9 (Highlight): "It's a really simple interface that allows me to focus on the speech and not on remembering the complex rules and definitions of emotions. Only remembering the last sentence is not hard, at least for me."* **Participants also mentioned**

the continuum and relative labeling interface design reducing oneself's variations and errors. *P17 (Relative): "It helps to find out and suppress the variations." P10 (Highlight): "It was so easy for labeling to reduce identifying errors when in a flow. Otherwise back and forth can be annoying." P2 (Highlight): "It might reduce the variation reported because of the flow between turns. It makes you ignore or average the variation within a turn."*

*Intuitiveness.* There is a statistical significance between **Highlight** vs. **Absolute** ($p < 0.05$) and **Relative** vs. **Absolute** ($p < 0.05$). **Most reasons were that up key and down key were according to the emotion change.** *P4 (Highlight): "Using the up and down makes sense when labeling certain things." P9 (Highlight): "The up and down arrows are intuitive to use and easy to remember." P15 (Relative): "It just matches well - up for more of an emotion, down for less of an emotion." P19 (Relative): "I think it's pretty simple to assume the up arrow means more and the down arrow means less. Same with the forward/back arrows."*

*Helpfulness.* There is a statistical significance in **Highlight** vs. **Relative** ($p < 0.05$) and **Highlight** vs. **Absolute** ($p < 0.01$). **Participants mentioned that highlights helped confirm their decisions.** *P3 (Highlight): "It helps to confirm my labeling." P10 (Highlight): "Make us sure that the reaction is going to change." P8 (Highlight): "Help people that do not feel emotion know what emotion changes should sound like."* **They also mentioned that highlights helped lower mental effort.** *P1 (Highlight): "Yes, the highlight is instrumental to completing the task with further support and I take those sentences more seriously." P7 (Highlight): "I definitely think the yellow helped because it alerted me to the system's labels and made me focus more on those sections." P12 (Highlight): "Yes because I could concentrate on the part I have to judge."*

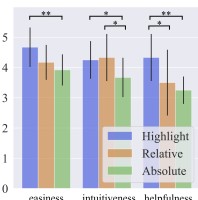

Figure 10: Questionnaire quantitative results.
(*: $p < 0.05$, **: $p < 0.01$).

Participants also mentioned some advantages of the interface and their labeling experience. **Many mentioned the "quickness" in the non-stop playing and keyboard interaction design.** *P3 (Highlight): "The direction keys made it very easy to use. I could make my selections quickly." P7 (Highlight): "It will allow quick and accurate annotations." P17 (Relative): "This helps people that might be slower than others in making their choices."* As for "accurate annotation" mentioned by P7, our understanding is that the participant just felt that keyboard operation was more reliable than mouse-clicking on the graphical interface. **There is a main disadvantage from Absolute group not found in the Highlight and Relative group, wanting more practice and tutorials.** *P27 (Absolute): "The tool was fun to use but I feel like more tutorials on how to rate it would help with becoming more proficient with the use of this tool." P31 (Absolute): "I think the tutorial is not sufficient for understanding the rating rules." P34 (Absolute): "The practice is helpful for the operation but not for how to rate emotions. I gave the rating based on my feelings."* This means that our interface design gives rise to a short learning curve when compared to the traditional interface.

Meanwhile, there are also limitations pointed out by participants. Two mentioned that highlights were different from their decision. *P11 (Highlight): "Most sentences had emotion changes and it would not highlight them." P4 (Highlight): "To be honest, I find the high-*

*light sometimes didn't sit well with what I personally felt."* Another unignorable limitation is that there is cross-talk in the audio, which makes the task difficult and confusing. It is mentioned in all groups. *P1 (Highlight): "I think it can be difficult to compare one speaker's emotion between sections when the other speaker talks for a while in between." P15 (Relative): "There was a bit too much cross-talk that the system didn't deal with." P27 (Absolute): "Several times, there were parts in the audio when two speakers were speaking and it was hard to concentrate on the speaker to label."*

## 5.6 Results Discussion

We answer the Research Questions raised at the beginning of this section, and give an insight into what we found in the user study. Our approach can improve the annotation agreement and elicit valid and reliable labels from non-expert annotators. It also improve the user experience according to the questionnaire.

**Relative labels improved the annotation agreement and highlight suggestions further improved it (RQ1)**. As results shown in Sect. 5.5.1, we found the percent agreement of **Highlight** improved as compared to **Relative** and then **Absolute**. We also considered the agreement expected by chance, and we found **Highlight** exceeded the standard of quality of both Krippendorff's Alpha and Fleiss' Kappa in arousal. We did not find the same results in pleasure. We interpret it as that different cultural backgrounds and experiences lead to the chance agreement on pleasure [1], as people understand the pleasure difference in different ways.

We also noticed that participants found highlights were not perfectly consistent with their perceptions. We admitted the fact that the SER network did not perfectly predict highlights. However, after a deeper investigation of the percent agreement between correctly predicted sentences and incorrectly predicted sentences, no significance was found between them. Based on these results, we conclude that even though the network performance in this user study might not be perfect, participants could still reach an agreement, even though it interfered with the annotation accuracy of some participants to some degree.

***RelRoll* improved the annotation accuracy, and the relative labels elicited by *RelRoll* were valid (RQ2)**. Results in Sect. 5.5.2 shows that **Highlight** successfully elicited labels of sufficient accuracy from non-expert annotators. In contrast, **Absolute** failed to get labels of sufficient quality, even lower than the neural network prediction. We believe that it is because non-experts could not give reliable absolute annotations when first using the traditional absolute labeling interface.

**The interface of *RelRoll* was easier, more intuitive, and more helpful for non-experts, which lowered the learning curve of annotation (RQ3)**. Results in Sect. 5.5.4 prove the improved user experience of **Highlight**. From the questionnaire, we also found the learning curve of annotation was lowered. Learning annotation is profound because participants need to learn the relationship between numerical values and emotions in an absolute interface. As for the absolute interface, participants need to learn the full range of relationships between numerical values and emotions, which can be infeasible [1]. Meanwhile, highlights would allow users to only focus on the most possible sentences whose pleasure or arousal changes. However, the learning effect is also found in **Highlight** condition. More time and more rewind counts were found in the beginning tasks. We believe this is because (1) users needed to learn the indication of highlights by reviewing their decisions and (2) users were not confident with the highlight predictions when first using the interface.

## 6 LIMITATIONS AND FUTURE WORK

We discussed about the limitations in *RelRoll* and potential future directions in this section.

*Dependence on experts' annotations.* *RelRoll* is a mechanism of eliciting reliable annotations by combining experts and non-experts. This approach tries to make annotations by non-experts following the distribution of experts' annotations. Thus it is vital to select representative and reliable data annotated by experts as *training set* before using this mechanism. In the future, we plan to find an optimized method to select the data for the *training set*.

*Nuances in design decisions.* Even though we validated *RelRoll* through a user study, there are nuances in design decisions we have not validated. There can be different designs on the number of levels of relative labels, which can be interesting to explore. In addition, we chose one approach to mapping between absolute differences to relative labels. There could be other rules to do the mapping depending on practical use cases.

*Mechanism scalability.* It may become infeasible for to use *RelRoll* annotate all emotional dimensions by playing an audio recording only time. However, it is related to participants' expertise in emotion understanding. One possible solution is to predict other dimensions when non-expert participants annotate one specific dimension. In the user study, we tested only one 2 minutes 40 seconds conversation. However, if removing overhead, we could acquire three times more than the current results. An extra user study can be conducted to prove the validity of a larger dataset.

*Dealing With Cross-Talk.* Participants from the user study mentioned that cross-talk influenced their understanding and decisions. Since our mechanism requests users to focus on one speaker in each task, it is important to reduce the effect of cross-talk. However, cross-talk can be common in spontaneous speech, which refers to unsmooth or inappropriate *turn-taking* in linguistics, even though sometimes they are considered natural or unnoticeable for speakers. Voice separation can be applied to extract the speech by the speaker to label when there is a mixed speech of multiple speakers talking at the same time. It will reduce users' difficulty in understanding the speech and introduce a better experience.

*Other Future Directions.* It is not supported for users to express uncertainty (e.g., unsure about labels) in the current interface design. It is interesting to consider and quantify the uncertainty in future work. There can be more applications of *RelRoll* on other scoring annotation tasks, such as image quality assessment, speech intelligibility rating, etc. In addition, there can be more applications for different data formats, such as videos, streams of images, etc. Last but not the least, while *RelRoll* can be effective on continuous data like speech, it can be difficult to adapt it to discrete data. One possible solution is grouping the discrete data into a sequence of an appropriate order before presenting them to users.

## 7 CONCLUSION

This paper presents a mechanism allowing non-expert annotators to give reliable scoring annotations. The key idea is to reduce the labeling difficulty by suggesting changes and estimating absolute values from relative labels. User study results showed that the interface of *RelRoll* can improve the inter-annotator agreement and obtain reliable and valid labels. In addition, it can improve non-experts annotation accuracy as compared to the absolute annotation interface. We also found that this interface can leverage the user experience from the quantitative results and qualitative feedback. We hope *RelRoll* can inspire more research on different scoring annotation tasks other than speech emotion annotation, which can involve more non-experts in originally expert-oriented labeling tasks.

## ACKNOWLEDGMENTS

This work was supported by JST SPRING, Grant Number JP-MJSP2108.

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
