# OpenReview forum: "RelRoll: A Relative Elicitation Mechanism for Scoring Annotation with A Case Study on Speech Emotion"
_graphicsinterface.org/Graphics_Interface/2023/Conference — GI 2023_

### Official Review · Reviewer_5zWA · 2023-01-06
**solid work but with some flaws in stats analysis**

**Rating:** 9
**Confidence:** 4

**Review:**

This paper presents a new annotation system for speech emotion using a relative elicitation method. This paper definitely addresses a very important problem as labelling is essential in training high quality machine learning models. The paper is well motivated and well written. In the introduction, the part “Instead of learning the full range …” may needs more explanation and grounding, while this is expanded later. Right now it is unclear why relative estimation is considered easier. Also, the paper can be benefited by better defining experts and non-experts. These are not clear in the introduction. Who are considered experts and who are not? What are the specific users that this tool is designed for?

The related work reads mostly fine. It needs to better distinguish the proposed technique from the closest work such as SCRIBE. The following papers might be of interest:

[1] Felix et al. 2018. The Exploratory Labeling Assistant: Mixed-Initiative Label Curation with Large Document Collections. Proceedings of the 31st Annual ACM Symposium on User Interface Software and Technology (UIST '18), 153–164.

[2] Shen et al. 2022. OLALA: Object-Level Active Learning for Efficient Document Layout Annotation. Proceedings of the 5TH Workshop on NLP and Computational Social Science at EMNLP.

The system UI is well designed with simplicity and the method is clearly described. The user study design is rigorous. However, the testing data, 2min 40s conversation sounds too short, and only one conversation is used in the study. The authors want to discuss this limitation and the generalizability of the results because of a small dataset is used. The results are nicely presented. However, there is one important issue. The authors should redo the analysis on the questionnaire results. Questionnaire ratings are ordinal data and non-parametric tests should be used (e.g., Friedman test) instead of ANOVA. Note this may change some of the significant results and conclusions.

Overall, I think this is a strong paper but has some issues to fix, especially for the analysis of the results. This may need a major revision but should be done within the conference review cycle.

---

### Official Review · Reviewer_j2QC · 2023-01-14
**The authors created a visual interface called RelRoll to enable non-experts to provide relative labels to speech emotion data. The authors provide a rationale for relying on relative labels and the design choices that they make for RelRoll, in addition to a crowdsourced evaluation.**

**Rating:** 8
**Confidence:** 4

**Review:**

The authors tackle a very important problem in machine learning, namely, labeling issues which can cause significant problems in building robust machine learning models. The authors designed and implemented RelRoll, an interactive tool to enable non-experts to provide relative labels to speech emotion data.

The authors address a valid and longstanding problem in the machine learning community, by focusing on providing non-expert labelers with the ability to provide labels. The visual interface is simple, intuitive and builds upon familiar visual metaphors that are used in presenting audio data.

The authors evaluated their approach with an MTurk study with 36 non-expert labelers, using RelRoll in three different configurations - Highlight (using a visual marker (based on neural network generated suggestions) to represent strong changes), Relative (enabling annotators to provide relative labels) and absolute (asking users to provide absolute labels). The user study found improvements over the state of the art through RelRoll.

Overall, the paper is well written and the results are clearly presented. The authors could add more detailed discussion about the limitations of their approach by also including details on the scalability of such an approach, since it may become infeasible to annotate recordings one at a time. In addition, the authors can also discuss other design choices and alternatives that can be implemented in the future, such as encoding additional metrics (apart from the simple encodings of high and low emotional changes) etc.

---

### Official Review · Reviewer_Wo41 · 2023-01-16
**Decent contribution, but needs clarification on motivation and implementation details; these can likely be handled with editorial oversight**

**Rating:** 6
**Confidence:** 3

**Review:**

Summary
This paper proposes a new system to enable non-experts to label affective states more accurately and consistently, using speech emotion as a target domain. The system has three different interfaces: the status quo (an absolute rating system for arousal and pleasure, which typically requires trained experts which can be hard to recruit and expensive), a relative rating system (where participants just label whether a speech excerpt is much higher, higher, the same, lower, or much lower than the previous one), and a relative rating system w/ a highlight (where a neural network predicts which speech elements change vs stay the same). These three interfaces are compared with a between-participants study (N=12 per condition) using Mechanical Turk, and show that Highlight performs best on rating agreement measures, accuracy with a ground truth, and usability ratings; Relative is next, then absolute.

Review
I argue for accept, if clarifications are made. I believe this paper is interesting, useful, and (to my knowledge) novel. The ability to enable non-experts to label data does increase the ability to label affective data, and the study seems quite conclusive in its comparisons. Overall, I think that this paper is strong enough to accept, but will need several improvements for camera ready. I think these can be evaluated without follow-up peer review, i.e., confirmed by an editor, so I argue for accept. (In journal-speak, this would be minor revisions without follow-up external review.)

However, the presentation suffers and several improvements need to be made to the paper to make sure it is replicable and can be evaluated by readers. The motivation in the abstract and introduction is difficult to understand at times, and it took a careful read-through of the paper to understand what exactly the authors were implementing and why (highlighting, for example, comes out of nowhere during implementation). Implementation details are difficult to understand and missing - the neural network equations do not explain all terms nor do they explain some aspects (e.g., what kind of regularization was used?), terms aren't always unpacked ("SER" network?), and the three interfaces are not explained adequately. I believe these clarifications are necessary before it can be published.

Here are a list of additional questions and comments:
 - "Difficult scoring annotation", the three first words, is clumsy and confusing. The introduction could be improved upon to be clearer
 - "a 36 people" study on page 2 reads awkwardly
 - Related work seems appropriately covered, although I am not expert in this area
 - Equations 4 and 5-7 are confusing and not always explained, I believe this section needs substantial revisions
 - Fig 5 should explain each of the different conditions better, i.e., how each of them works. A new subsection should be added to motivate these, especially because Highlighting is a major design effort but has no rationale that I can see.
 - Sections 5.4 and 5.5 is a little confusing but generally valid

---

### Meta-Review · Area_Chair_QWQs · 2023-01-18

**Recommendation:** 8
**Confidence:** 5

**Metareview:**

I recommend to accept this paper. We have three reviews, all of which recommend for acceptance with varying levels of support.

This paper has a strong contribution, tackles an important task, describes a sound study, and is well written. It would be a clear contribution to GI 2023.

That said, all reviewers pointed out improvements that can be made with this paper, including revisions to the statistical analysis and improvements to related work, improved discussion of limitations with the technique (e.g., scalability), and improved clarity of motivation and implementation details. Given the positivity of the reviewers, I believe these can be completed in a camera-ready revision.